# Supermoms—Tired, admired, or inspired? Decoding the impact of supermom beliefs: A study on Indian employed mothers

**Shalaka Sharad Shah**[1*☉], **Smita Chaudhry**[2☉], **Shilpa Shinde**[3☉]

**1** Department of Psychological Sciences, School of Liberal Education, FLAME University, Pune, India,
**2** Department of Human Resources & Organizational Behaviour, School of Business, FLAME University, Pune, India, **3** Department of Human Resources & Organizational Behaviour, School of Business, FLAME University, Pune, India

☉ These authors contributed equally to this work.
* shalaka.shah@flame.edu.in

## Abstract

Some days she is a supermom, some days just a mom, but most days a bit of both, and every day she strives to keep her oath! The study aims to unravel the perceptions and beliefs about 'being a supermom' and to explore its relationships with work-related factors. Our study aims to unravel the perceptions and beliefs of 306 Indian-employed mothers about 'being a supermom' and its relationship with work-related factors like self-efficacy, work engagement, and psychological well-being. Standardized scales were used for quantitative study and qualitative questions were used to understand the beliefs about the supermom notion. Frequency analysis for the perception of the supermom notion showed that 52% of mothers consider it detrimental to achieving success, while 48% consider it to be beneficial. A path analysis revealed that if the supermom notion is perceived as beneficial, it increases employed mothers' self-efficacy; self-efficacy promotes work engagement, and work engagement increases psychological well-being. SEM confirmed that self-efficacy increases psychological well-being directly as well as through work engagement indicating partial mediation of work engagement. Findings suggest that while the supermom ideal can be exhausting and is often viewed as a myth or trap that negatively impacts psychological well-being, it can also serve as a source of inspiration and contribute positively to psychological well-being for some mothers.

## Introduction

The changes in society at large and workplaces in general have led to more and more women choosing to play the dual role of breadwinner and homemaker [1]. This amalgamation of self-identities has resulted in women having to prove that they can actually 'do it all' [2] while playing multiple roles in society [3]. Although a strong identification with the dual role has been found to have a positive relationship with higher well-being, a perceived identity conflict between these roles has been found to have a negative relationship with employed mothers' well-being [4]. A popular notion that may influence how employed mothers perceive and reconcile these dual roles is the notion of supermom. A broad definition of the supermom

**Data availability statement:** Shah, Dr Shalaka (2024), "Data of Indian Mothers", Mendeley Data, V1, DOI: . https://doi.org/10.17632/2jtx5864yx.1

**Funding:** The author(s) received no specific funding for this work.

**Competing interests:** The authors have declared that no competing interests exist.

notion states that "being a supermom demands mothers to have higher (super) capacities to be able to perform well in all life domains. In a practical sense, a supermom has a job, can smoothly carry out household/family responsibilities, can present herself in full control, and can keep herself together and is on top of it all" (p. 9) [3]. Furthermore, the notion of the supermom refers to women who navigate both public and private spheres, implying that this balancing act demands superhuman capacities [5]. The notion of supermom is strengthened by other societal myths and ideals. Researchers explain the ill effects of the notion of 'The New Momism' [6] and how it has been oppressive towards both stay-at-home and employed mothers. This notion of 'New Momism' which is termed a myth begins with the insistence that a woman must be devoted to the care of her children 24/7 to be considered a decent mother. It has led to stay-at-home mothers trying to do intensive mothering and employed mothers trying to show that they can be supermoms, managing to work well in jobs while raising happy children. Similarly, Indian cultural symbols such as multi-handed Goddesses of power, learning, wisdom, wealth, and peace are worshipped, encouraging women to believe in the ideal of a 'superwoman' whose innate ability helps them to excel in multiple roles [7]. These ideals prevalent in the Indian culture shape the meaning women attach to motherhood. According to the symbolic interactionism framework, in a society, social interaction influences the development of self-identity [8].

Although challenges for employed mothers in India have become increasingly diverse [9,10] research has found that Indian society largely puts familial and childcare responsibilities on women rather than men [9]. Such social and cultural expectations endorse the ideals of supermom. Past research has indicated how motherhood and employment can have a combined impact on the well-being of mothers due to the associated challenges as well as the mental load of managing multiple roles [11–14]. This study is aimed at adding to the literature by exploring the relationship between the perceptions and beliefs of employed mothers about the supermom notion. Well-being can also be affected by one's belief about how effectively they can perform a task, i.e., self-efficacy [15–17] and how involved one is in their work, i.e., work engagement [18–21]. Beliefs about the supermom notion may influence the self-efficacy and work engagement of employed mothers, and therefore their psychological well-being. To the best of our knowledge, these relationships have not been examined in existing studies. This study seeks to address this gap. The study focuses on mothers who are formally employed in organizations. As a recognition of women's invisible labor [13] and work (childcare, housework, emotional care-giving) that women tirelessly engage in, irrespective of whether they are homemakers or employed, we have consciously avoided the term 'working women' and instead chosen the term 'employed mothers', to describe the target population of our research.

## Literature review

### Understanding motherhood and its implications for women in the Indian context

**Negative consequences of motherhood.** Most employed mothers predominantly play two significant roles - at home and in the workplace. At home, the roles become multiple - a mother, a wife, a daughter-in-law, a sister, a sister-in-law, apart from roles concerning other relatives. Therefore, the notion of motherhood goes beyond child-rearing extending itself into these other familial roles. This stereotypical ideology of motherhood is a reality in almost all cultures across the globe [22]. It affects the mother's mental and physical health [23–25] as pursuing a career does not absolve employed mothers from playing these multiple roles [26]. Employed mothers may face negative outcomes of perceived neglect of the motherly

roles, leading to maternal guilt [27] as well as stress and anxiety [28] while simultaneously experiencing a pressing need to prove themselves outside the home.

More recent work in this area uncovers women's fear of becoming mothers and explains that it is due to the notion of 'good mother' nested in the moral context of motherhood [29]. It is therefore not surprising that there is evidence of a negative association between the presence of small children and urban married women's employment [30,31]. Juggling the simultaneous demanding needs of child rearing and a job places a penalty on almost all employed mothers [32] unless organizations provide formal or informal support systems for childcare [33,34]. It is found that in India too lack of child support severely affects women's ability to work [35]. To add to this hurdle, employed mothers face discrimination at the workplace, due to the social perception that they will be less efficient and may discontinue their work to bring up their children [36–38].

**Positive consequences of motherhood.** However, as every cloud has a silver lining, motherhood is also viewed positively. It works as an alternative source of female identity [22]. Motherhood can be partially rewarding in female-led households when it allows women some power to influence their children's well-being [39]. Due to the changes in the economy, there is an increased participation of mothers in the workforce [10]. This helps women who are also mothers contribute to household expenses [40] as well as experience higher self-confidence in managing work and family [41].

**Psychological well-being and its relation to motherhood.** Health is defined as a state of complete physical, mental, and social well-being, and not only the absence of disease [42]. Mental health is defined as a state of well-being in which an individual realizes their abilities, can cope with normal life stressors, can work fruitfully and productively, and can contribute to their community [43]. Mothers' psychological well-being is influenced by several factors - employment status being one of them. The research focusing on multiple roles performed by women has shed light on the association of social roles and social relationships with psychological well-being [4,44]. Some women feel bogged down by numerous roles and some flourish [45,46]. These effects of multiple social roles are understood using two perspectives - the role enhancement perspective and the role strain perspective.

Although engaging in multiple roles has been found to have negative outcomes in some studies, as it creates a strain on the individual or even a conflicting situation in many instances researchers have argued that it is not the workload itself, but the degree to which individuals can exercise control over themselves that largely determines the effects of multiple roles on their accomplishments [47,48]. The role strain perspective suggests that engaging in multiple roles reduces overall well-being as it causes role overload leading to role conflict [49–52]. The multiple roles create strain for mothers [53] in the form of reducing their well-being [14,52,54] reducing their chances of career success [55] and leading to experience of discrimination in terms of wages [56,57].

On the positive side, role enhancement theory proposes that playing more than one role provides multiple resources and support systems empowering individuals to better cope with role demands [58–61]. Multiple roles can satisfy psychological needs [61], and positively affect one's health [62] and psychological well-being [63]. Research shows that children provide not only happiness and a sense of accomplishment to women; the process of coping with multiple demands may lead to an increase in competencies [53,64] as well as emotional intelligence to the extent of preparing them for senior positions at work [65].

**The notion of supermom: positive and negative perspectives.** We believe that the supermom notion may be perceived positively or negatively depending on individual mindset shaped by personality and social experiences and interactions. A supermom is the one who seamlessly manages a career and household responsibilities, and motherhood has

been idealized, particularly in urban, middle-class India [66]. This portrayal reflects how societal norms, reinforced through media, family structures, and cultural values, contribute to gendered expectations by presenting multitasking as an inherent capability of women [67].

Although the notion can be studied from either perspective, we were interested in exploring the positive beliefs about the supermom notion to understand whether it deserves to be endorsed the way it is currently influencing our culture and society. Our study aimed to examine the impact of the positive beliefs about the supermom notion (i.e., supermom as a beneficial notion) on self-efficacy, work engagement, and psychological well-being of employed mothers.

**Theoretical rationale for the supermom notion and its relation to selected variables.** A positive perception of the supermom notion by employed mothers may develop due to their confidence in their ability to balance the dual role of a mother and employee. According to the Role Balance Theory [68], one can create a positive role balance between the multiple roles in their total role system by being fully engaged and paying full attention to every role. This positive role balance is positively associated with a sense of ease and well-being. The positive beliefs about the supermom notion align with the Role Enhancement theory, which suggests that employment benefits women by expanding their emotional and economic resources. These enhanced resources can help mitigate the challenges of managing multiple roles [58]. One outcome of the beliefs associated with the positive notion of supermom would be improved self-efficacy. As stated by Self-efficacy theory [47], one's perceptions of personal ability, knowledge, skills, past accomplishment of tasks as well as experience of positive or negative emotions influence the development of self-efficacy. Consequently, perceptions and beliefs of the supermom notion can shape self-efficacy, a dynamic construct that evolves. Higher self-efficacy is expected to influence well-being positively as several studies relate self-efficacy positively to a range of positive outcomes including successful work-related performance [69,70]. The Conservation of Resources theory considers self-efficacy as a psychological resource. Studies show that individuals with high self-efficacy can gain, maintain, and conserve resources [71]. These resources help to deal with challenges experienced in both personal and work life allowing individuals with high self-efficacy to experience work engagement [69]. Further, work engagement as a positive motivational-affective state leads to the experience of positive emotions. Fredrickson's Broaden and Build theory explains the role of positive emotions in leading to higher psychological well-being [72].

We thus hypothesized that supermom as a beneficial notion has a positive relation with self-efficacy, self-efficacy has a positive relation with psychological well-being and work engagement, and work engagement further has a positive relationship with psychological well-being. The proposed hypotheses have been presented in Fig 1. We have hereafter referred to the positive notion of supermom as supermom (beneficial) in the paper.

**The notion of supermom and self-efficacy.** Self-efficacy refers to people's belief about their ability to do a task effectively [47] and is defined as "people's judgments of their capabilities to organize and execute courses of action required to attain designated types of performances" [73] (p. 391). Studies have explored the relationship between handling multiple roles and self-efficacy. A study concluded that various roles were perceived as challenging by some employed women who may thrive on them rather than feel overwhelmed by them. They found that employed women had higher levels of generalized and domain-specific self-efficacy than non-employed women, influenced by their belief in managing the multiple demands and involvement in work and family [45]. We thus argue that employed mothers who perceive the notion of supermom as beneficial can manage multiple roles, and this may result in high self-efficacy in carrying out these roles. Therefore, we propose the following hypothesis (see Fig 1).

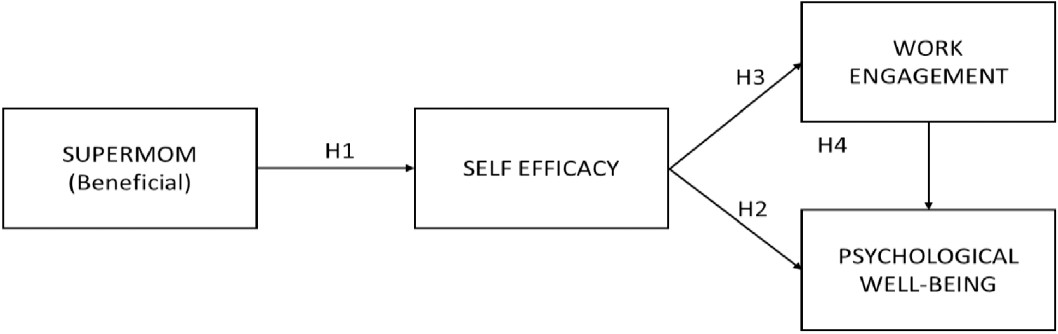

**Fig 1. Conceptual model for the relation between supermom (beneficial) & psychological well-being.**

H1: Supermom (beneficial) has a positive relation with the self-efficacy of employed mothers.

**Self-efficacy and psychological well-being.** Higher levels of self-efficacy have several benefits for an individual. It not only protects them against distress and illnesses [74] but also contributes to a sense of positive well-being for an individual [16]. Self-efficacy beliefs have also been known to predict performance [70,75]. Another study found that occupational self-efficacy positively influenced employed women's career aspirations [46]. High self-efficacy levels are also likely to increase intrinsic interest in an activity and lead individuals to become absorbed by it [47,76,77]. The construct of psychological capital includes self-efficacy as one of the four components, which has been found to have a positive relationship with psychological well-being in employees in general [78,79] and in women employees [80].

An optimistic view of self-efficacy is a solid foundation for accomplishments and positive well-being. Individuals need to feel efficacious to consistently put in effort to accomplish daily tasks as they are accompanied by seen and unforeseen hurdles. Psychological well-being and accomplishments are possible due to this affirmative sense of self-efficacy. Conclusively, self-efficacy is related to health-promoting positive feelings [45]. Therefore, we propose the following hypothesis (see Fig 1).

H2: Self-efficacy is positively related to the psychological well-being of employed mothers.

**Role of work engagement.** Work engagement is defined as a "positive, fulfilling, work-related state of mind characterized by vigor, dedication, and absorption. Rather than a momentary and specific state, engagement refers to a more persistent and pervasive affective-cognitive state that is not focused on any particular object, event, individual, or behavior" [81] (p.74). Some researchers have compared engagement with the phenomenon of 'flow' [82]. They discuss engagement in terms of the positive feelings that employees experience when completely engaged, similar to the experience of flow. Work engagement has become an important phenomenon to study, as research has shown a positive relationship between work engagement and performance [81,83], creativity and productivity [83], work engagement, and occupational well-being [84].

Several studies have also discussed the relationship between self-efficacy and work engagement. Research on restaurant employees in Greece found that certain job resources contributed to personal resources of optimism, self-efficacy, and self-esteem daily, leading to greater engagement in work and better performance [85]. A study on employees of an electronics company in the Netherlands tested long-term relationships between job resources, personal resources, and work engagement, and found that self-efficacy was one of the personal resources that has a positive relationship with work engagement [86]. Another study found that

individuals high in self-efficacy experienced higher levels of flow [81] which has been related to work engagement as discussed above. There is evidence explaining how task mastery, one of the sources of self-efficacy, would lead to the absorption dimension of work engagement as employees would be able to understand the detailed steps of the task easily [87].

Considering the literature presented, we propose that the self-efficacy of employed mothers would lead to their work engagement. Although we could not find any contextual studies exploring this relationship, we propose that employed mothers who feel capable of doing their work well would also be more interested and involved in their work because they are confident of performing well and getting commensurate results. Thus, we proposed the following hypothesis (see Fig 1).

H3: Self-efficacy is positively related to the work engagement of employed mothers.

Current literature has also discussed the effect of work engagement on psychological well-being. A study on women managers and professionals working in a large Turkish bank found that work engagement was positively associated with indicators of psychological well-being such as low levels of exhaustion and psychosomatic symptoms [18]. In another study on nurses in Turkish hospitals, work engagement was shown to have a consistent and moderate relationship with indicators of psychological well-being [88]. Therefore, we argue that work engagement can increase psychological well-being.

However, it was found that burnout, which is at the other end of the continuum of work engagement, is associated with decreased work engagement, and lower psychological well-being [89]. Specifically, in the case of employed women, studies have found that when they dislike the content and context of their work, they do not gain in terms of psychological health [90]. Considering H2 and H3 mentioned earlier, we argue that the positive effect of self-efficacy on work engagement would lead to this relationship positively impacting psychological well-being. Thus, we argue that self-efficacy would increase psychological well-being directly and indirectly by increasing the work engagement of employed mothers. Therefore, we propose the following hypothesis (see Fig 1).

H4: Work engagement partially mediates the relation between self-efficacy and the psychological well-being of employed mothers.

## Research methodology

We employed an embedded/nested mixed methods design that allowed us to collect qualitative and quantitative data simultaneously and conduct analysis [91]. The design involved an online survey with dominant quantitative questions and a few qualitative questions embedded in them. Qualitative data pertained to the respondent's perceptions and beliefs about the supermom notion. Quantitative data pertained to respondent's demographic information and self-evaluation of self-efficacy, work engagement, and psychological well-being. This embedded/nested mixed-methods design explored respondents' perceptions and beliefs about the supermom notion qualitatively followed by a quantitative examination of its relationship with self-efficacy, work engagement, and psychological well-being. This was required as we expected differences in employed mothers' perceptions and beliefs about the supermom notion which would in turn affect how they felt about the notion and its utility in achieving success in work and non-work life domains. We obtained approval from the host University's Institutional Review Board to conduct the research (Approval Number: 2024/04/02/EXP) and the data was collected directly from employed mothers after noting their written consent in the data collection Google form. Before the survey questions began, participants had to read the information about the study and provide their written consent by responding either "Yes" or "No" to the statement "I consent voluntarily to be a participant in this study" included in the Google Form.

## Sample selection

Considering the purpose of the study, we collected data through convenience and snowball sampling. The potential participants for the study were contacted through the authors' personal and professional networks. We assured them about the confidentiality and anonymity of their responses. After getting their written consent on the Google Form, we collected data from them. Using snowball sampling, we collected data from other employed mothers, whose contact information we received from our initial participants.

Our target participants were women between the ages of 25 and 50 years, who were well versed in the English language, mothers of a child/children, engaged in full-time or part-time paid employment in any organization, or were self-employed. They could be married, partnered, divorced, separated, or single mothers. All participants were Indian residents currently working and based in India.

## Sample description

Considering that the sample size had to be more than 5 times the number of items in the survey [83], we needed at least 280 responses. We could collect complete data from 322 respondents to meet the sample size requirement. About 47.4% of the respondents were in the age group of 45–50 years. About 94% percent of the respondents were married. About 56.5% had 2 children and 41.5% had one child. 49.7% had a post-graduation degree and 80.4% had full-time employment. 89.9% worked in the private sector. The average working hours for them in a week were 43, and the average number of working days was 5.56 in a week. 33% of respondents had spent up to 5 years in the current organization and 34% had a total work experience of more than 20 years.

The respondents represented industry sectors like education (schools, colleges, universities) (35%), manufacturing, start-up, business (22%), information technology (10%), art, fashion, media, marketing, travel & tourism (10%), medicine (doctors), pharmaceuticals (8%), banking, finance, insurance (7%), health, mental health, fitness (5%) and hospitality, aviation (3%).

## Survey design

An online survey was created to collect qualitative and quantitative data using a Google Form and data was collected from 14.05.2024 to 31.05.2024 after the IRB approval. The online survey included three open-ended questions (qualitative), items from established psychological scales measuring self-efficacy, work engagement, and psychological well-being, (quantitative), and some closed-ended questions on demographics. No identifying information was collected through the online survey. The open-ended questions explored respondents' perspectives regarding the supermom notion. An initial pilot study was conducted with 10 respondents resulting in satisfactory answers to the qualitative questions. This strengthened our confidence that these responses would offer meaningful insights into the perceptions and beliefs about the supermom notion. Based on the findings from the pilot study, the online survey questions were finalized. The participants from the pilot study were excluded from the final study. All survey questions were presented in English.

## Qualitative method

We used the methodological framework of symbolic interactionism for formulating the qualitative questions. This method focuses on the subjective meanings individuals assign to their actions and surroundings [92]. This perspective assumes that people shape their perceptions and interpretations through interactions with others. The core idea of symbolic interactionism is subjective meaning. Individuals "create shared meanings through their interactions, and those meanings become their reality" [93]. Three core principles are outlined in this approach

[94]. First, people act toward things based on the meanings those things hold for them. Second, these meanings are derived from social interactions with others. Third, individuals interpret and modify these meanings through personal engagement with their experiences. Together, these principles emphasize the importance of exploring how individuals attribute meaning to objects, events, and experiences. We believe that the perception and belief about the supermom notion among our respondents would be influenced by these three principles. Reconstructing these subjective perspectives serves as a valuable method for analyzing social realities [92] (p. 67) [95]. Symbolic interactionism has been widely applied across fields like sociology, anthropology, criminology, psychology, and education [95–97].

Qualitative data was collected using three open-ended questions embedded in the online survey. The questions were intentionally kept broad to capture participants' perceptions of the supermom notion and examine whether they believe this notion contributes to their success in work and non-work life domains. The formulated questions are: 1) Who is a supermom according to you? 2) Do you agree that you have to be a supermom to achieve success in all domains (work, family, life)? 3). According to you, is being a supermom beneficial to achieve success in personal and professional life or is it a trap?

## Qualitative data analysis & results

Based on the methodological framework of symbolic interactionism [92], the responses to the questions addressing the subjective meaning of the supermom notion and its role in achieving success in work and non-work domains of life were thoroughly analysed. Qualitative analysis was done using content analysis [98] and frequency analysis. Responses to the qualitative questions uncovered perceptions and meaning about the supermom notion. Using content analysis [98], an inductive coding system was developed, and codes emerged from the textual units. The coding was done manually to avoid missing out on any nuances. This also allowed a systematic selection of relevant statements as a first step. Thus, focusing on relevant statements helped in deducing the meaning for Q1 (Who is a supermom according to you?), uncovering and interpreting the beliefs for Q2 (Do you agree that you have to be a supermom to achieve success in all domains (work, family, life)? And Q3 (According to you, is being a supermom beneficial to achieving success in personal and professional life or is it a trap?). The interpretative statements were then mapped with the questions asked in the survey. Some definitions of the supermom notion emerged from the content analysis of responses to Q1 and selected definitions are presented below.

> *"A supermom is the one who handles the office, home, children, family, friends, parents - everything right from morning to night and makes everyone happy including herself."*

> Indian navy officer and mother of two

> *"A supermom is a woman who successfully manages the household, cares for and creates an environment that is most healthy for herself, and her family (extended too) while holding a job or being active in her community constantly making an impact both towards people close to her and society in general."*

> Proprietor in education and mother of one

> *"A supermom is someone who is emotionally balanced and physically healthy even after job and home responsibilities."*

> Professor & HoD and mother of two

The definitions that emerged from the content analysis concur with the established definition of supermom which views it as a mother who has higher (super) capacities and can perform well in all life domains such as job, household/family responsibilities, can be in full control, keeping herself together and is on top of it all [3].

A deeper analysis of responses for Q2 and Q3 using the inductive coding system led to two main categories of perceiving the supermom notion - beneficial (coded as 1) and detrimental (coded as 0). The categories were derived from participants' responses, shedding light on their understanding of the supermom notion. Quotes reflecting positive feelings, perceptions, and outcomes where the supermom notion was believed to enhance performance in various life areas were coded as "beneficial perceptions". Conversely, quotes highlighting negative feelings, perceptions, and outcomes, such as feeling trapped by unrealistic expectations, pressure to be perfect, or losing individuality, were categorized as "detrimental perceptions". Participants with detrimental views saw no value in the supermom ideal and believed success could be achieved without adhering to it.

These categories of viewing the supermom notion as beneficial or detrimental by the participants can be justified using the Role theory [99–101] which highlights the concept of role-making, involving creating and redefining one's self-concept and roles as a guiding framework for social interactions [102]. The theory highlights the reciprocal nature of role relationships, where role expectations are understood in relation to counter roles. Therefore, participants who view the notion as enhancing their ability to succeed in multiple roles see the supermom notion as beneficial, while those experiencing role overload and performance pressure view it as detrimental. Table 1 provides selected quotes found in both categories.

To avoid bias in the coding process, we selected two students as coders. The students were provided a brief about the notion of supermom based on existing literature but were not made aware of the purpose of the study. The students coded the positive beliefs about supermom (beneficial) as 1, and the negative beliefs about supermom (detrimental) were coded as 0. We established inter-coder reliability which is recommended as a good practice in qualitative data analysis [103]. The percentage agreement between the coders was 95%. We selected all the responses that were given the same code by coders and removed the responses that were given different codes. Thus, from 322 responses, we arrived at 306 responses worth analyzing. The number of responses in each category is provided in Table 2.

Table 1. Selected quotes of responses from participants on beliefs about supermom.

| Respondent details | Quotes |
| --- | --- |
| **Belief about supermom as beneficial** | |
| A full-time teacher and a mother of three | *"According to me being a supermom is beneficial as the more you get involved in it the more you explore your life...these things make you perfect in all phases of life, personally and professionally."* |
| A full-time worker in Public Relations & Marketing and a mother of two | *"I feel all mothers are supermoms…the fact that in most cases they are responsible for new lives, bringing them up, teaching them, taking care of them, nurturing them…it's a superpower that women have been granted by nature. Any mother who is responsible for their children is a supermom. It takes a lot out of her - emotionally, physically, intellectually, and psychologically."* |
| A Vice President in the private sector and a mother of one | *"I am currently a supermom! It's beneficial to achieve success in personal and professional life as it completely changes your outlook towards life and teaches you to be more efficient in managing your time and to keep moving forward."* |
| **Belief about supermom as detrimental (myth or a trap)** | |
| A Vice President of an IT company and a mother of two | *"It's a myth- one definition of a supermom may be restrictive, and it depends on what an individual values. For me, it's a juggle between family and work, for someone else it may be their family-health."* |
| A full-time teacher and a mother of two | *"A mom is also a woman with dreams, desires, strength, and weaknesses. She deserves to be allowed to make mistakes and learn from them just like any other human being. And we won't expect any mistakes from a supermom, so it's a trap."* |
| A Vice President in Customer Service and a mother of one | *"It is a trap. The word supermom comes with a huge commitment and societal pressure to perform well. Standing up to the label itself is a challenge. However, every woman who can do well in each of their interested domains can be a supermom! Success is very subjective, and people can have their ways of achieving it."* |

**Table 2. Frequency analysis for the beliefs of the supermom notion.**

| Category | Code | Frequency | Percentage |
|---|---|---|---|
| **Supermom as Beneficial** | 1 | 148/306 | 48% |
| **Supermom as Detrimental** | 0 | 158/306 | 52% |
| | | **Total** | **100%** |

## Quantitative method

Since our conceptual model was based on the belief that supermom was beneficial, for which we did not have any quantitative measure, we defined a new numerical variable named SMB. SMB was a dichotomous ordinal variable with values 1 and 0. 1 denoted belief about supermom as beneficial, and 0 denoted belief about supermom as not beneficial. To populate SMB, we utilized the coding for the supermom notion for the 306 responses derived from the qualitative analysis. SMB carried a value of 1 for all responses from participants who viewed supermom as beneficial, and a value of 0 for all responses from participants who viewed supermom as a myth or a trap (thus not beneficial). We planned to use a structural equation model (SEM) and path analysis to test the model. Existing literature indicates that they can be used to analyse the relation of dichotomous ordinal variables (SMB) with continuous variables [104–106] if the continuous variables have a normal distribution for the two values of SMB.

We collected quantitative responses on self-efficacy, work engagement, and psychological well-being using standardized scales. Self-efficacy was measured using a 10-item scale Riggs et al., (1994) [107]. A 9-item Utrecht Work Engagement Scale Schaufeli & Bakker (2004) [89] was used to measure work engagement. Psychological well-being was measured on an 18-item scale Ryff & Keyes (1995) [108]. Although shorter versions of this scale (6–8 items) with improved psychometric quality are available [109], we went ahead with the original scale because it covers all the six dimensions of psychological well-being (sense of purpose in life, striving towards personal growth, environmental mastery, self-acceptance, autonomy, and positive interpersonal relationships) adequately [109]. We believe that all six are relevant in the context of the well-being of employed mothers.

The number of work hours a week was considered a control variable as it can determine an individual's well-being. Studies have shown that working hours can affect employed mothers' fatigue and stress [110]. Lower working hours can imply more quality time to spend with their children, and more time available to tend to their needs and wants and do household work. Mothers who believe in being supermom may have better psychological well-being if they have more time to engage in these activities. Studies have also found a relationship between flexible work hours (related to more hours at home) and lower stress levels, increased job enrichment, and improved job satisfaction and productivity [111].

## Quantitative data analysis & results

Although the constructs used in this study are well established, they have not been analysed in the context of employed Indian mothers. Hence, we performed validity and reliability analysis. We first conducted exploratory factor analysis (EFA) on the continuous variables to improve their validity by eliminating items having low factor loading (less than.05). We then conducted heterotrait-monotrait ratio of correlations (HTMT) to ensure the discriminant validity of these variables before we proceeded with further analysis. We then tested the reliability of the variables using Cronbach Alpha (α), and also calculated the composite reliability (CR). Our next step was to calculate inter-construct correlations using the Spearman method, to get an understanding of the relationship between the variables. We then performed confirmatory

factor analysis (CFA) to evaluate the alignment of the proposed model with the observed data. It also helped account for the measurement error in the observed variables, leading to a more accurate estimate of relationships between the variables. After obtaining satisfactory results, we tested the goodness-of-fit of the structural model and then used a path analysis on that model to test the hypotheses. We used IBM SPSS® and IBM AMOS® for the quantitative analysis. To ensure the usability of SMB in AMOS in a single group model, we tested whether the continuous variables met the normal distribution condition for the different values of SMB using multigroup analysis [112].

EFA led to removing one item from the work engagement scale and eight items from the psychological well-being scale. Consequently, 10 items of self-efficacy, eight items of work engagement, and 10 items of psychological well-being were retained. The HTMT analysis indicated the following: self-efficacy and work engagement (HTMT value = 0.423); self-efficacy and psychological well-being (HTMT value = 0.506); work engagement and psychological well-being (HTMT value = 0.487). As all HTMT values were less than 0.85, the discriminant validity of the variables was established [113]. Self-efficacy ($\alpha$ = 0.86, CR = 0.84), work engagement ($\alpha$ = 0.87, CR = 0.86), and psychological well-being ($\alpha$ = 0.80, CR = 0.78) also indicated acceptable inter-item consistency and composite reliability.

Given three predictor variables, a medium effect size level (.15), 95% confidence level ($\alpha$ =.05), and a power requirement of.80, G*Power indicated that the sample size of 306 was appropriate for testing correlation [114].

Considering that SMB and weekly hours are not continuous variables, we used the Spearman correlation between all variables (see Table 3). There was significant correlation of (a) SMB with self-efficacy ($r = 0.19$, $p <.001$), work engagement ($r = 0.11$, $p <.05$) and psychological well-being ($r = 0.20$, $p <.001$), (b) self-efficacy with work engagement ($r = 0.46$, $p <.001$) and psychological well-being ($r = 0.47$, $p <.001$), and (c) work engagement with psychological well-being ($r = 0.49$, $p <.001$).

To ensure that SMB could be used in AMOS under the maximum likelihood estimation, we did a multigroup analysis on SMB, and found that the continuous variables were normally distributed, for the fixed values of SMB [112]. CFA on the measurement model revealed a $\chi^2$ (351) = 527.44, p <.001. The chi-square value was high due to the use of a large sample size. To overcome this limitation, we used alternative fit indices that do not get affected by sample size yet indicate good fitment between the proposed model (i.e., population covariance matrix) and the sample data [115]. We used the absolute fit indices: CMIN/df, root-mean-square error of approximation (RMSEA) standardized root-mean-square residual (SRMR), and incremental fit index: comparative fit index (CFI). CMIN/df minimises the impact of sample size

**Table 3. Means, standard deviations, and intercorrelations.**

|   | Variables | M | SD | 1 | 2 | 3 | 4 | 5 |
|---|---|---|---|---|---|---|---|---|
| 1 | Week hours | 43.09 | 15.32 | 1 | | | | |
| 2 | SMB | 0.48 | 0.50 | 0.03 | 1 | | | |
| 3 | Self-efficacy | 6.01 | 0.70 | 0.11* | 0.19*** | 1 | | |
| 4 | Work engagement | 6.15 | 0.64 | −0.02 | 0.11* | 0.46*** | 1 | |
| 5 | Psychological Well-being | 5.85 | 0.69 | −0.01 | 0.20*** | 0.47*** | 0.49*** | 1 |

$N = 306$.

*$p < 0.05$;

**$p < 0.01$;

***$p < 0.001$. One-tailed significance values are reported.

and gives a more accurate and unbiased assessment of the fitment of the proposed model. RMSEA being sensitive to the number of estimated parameters, reveals the parsimony of a model alongside fitment to data. SRMR conveys the acceptability of the square root difference between the residuals of the sample and the population covariance matrix. CFI shows how well the proposed model fits the sample data compared to a baseline model having no relationships among variables. We obtained CMIN/df = 1.50, RMSEA = 0.04, SRMR = 0.05, and CFI = 0.94, indicating an acceptable fit for the measurement model [116,117]. The proposed structural model had $\chi^2$ (380) = 557.36, p <.001. Considering the sample size, we analysed alternative indices and found acceptable goodness of fit with CMIN/df = 1.47, RMSEA = 0.04, SRMR = 0.06, and CFI = 0.94. Thus, we could proceed with hypothesis testing.

## Hypothesis testing

Hypotheses were tested using a path analysis of the proposed model (Table 4). H1 stated that defining supermom (beneficial) positively relates to self-efficacy. Results show that SMB was significantly and positively related to self-efficacy ($\beta$ = 0.23, $p$ <.001). Thus, H1 was supported. H2 stated that self-efficacy positively relates to the psychological well-being of employed mothers. Results of the path analysis showed that self-efficacy was significantly and positively related to psychological well-being ($\beta$ =.39, $p$<.001). Thus, H2 was supported. H3 stated that self-efficacy positively relates to the work engagement of employed mothers. Results of the path analysis showed that self-efficacy was significantly and positively related to work engagement ($\beta$ =.49, $p$<.001). Thus, H3 was supported.

H4 stated that work engagement partially mediates the relation between self-efficacy and psychological well-being for employed mothers. The path analysis between work engagement and psychological well-being showed a significant positive relation ($\beta$ =.45, $p$<.001). These findings and the results of H2 and H3 suggest that self-efficacy may increase psychological well-being directly or indirectly through work engagement indicating mediation of work engagement. To confirm whether the mediation was full or partial, we compared the proposed structural model indicating partial mediation of work engagement, with an alternate model indicating full mediation (see Table 5). This comparison revealed that the proposed model fitted the sample data significantly better than the alternate model ($\Delta\chi2$ [1] = 27.81, p<.001) with a smaller $\chi^2$ and better alternative fit indices.

We additionally used the bootstrapping method [118] to test the indirect effects of self-efficacy on psychological well-being through work engagement (see Table 6). The results showed significance of all three effects (direct effect = 0.392, $p$ = 0.002; indirect effect = 0.221, $p$ = 0.001; total effect = 0.614, $p$ = 0.001). Thus, self-efficacy impacted psychological well-being directly, as well as through work engagement (indirect effect = 0.221, $s.e.$ = 0.056, 95% CI [0.133, 0.369]), confirming the partial mediating effect of work engagement. Thus, H4 was supported.

**Table 4. Path analysis.**

| Paths | $\beta$ | p-value |
|---|---|---|
| SMB to EF | .23 | <.001 |
| EF to WE | .49 | <.001 |
| WE to PWB | .45 | <.001 |
| EF to PWB | .39 | <.001 |
| WEEK_HRS to PWB | −.03 | .644 |

$N$ = 306. All coefficients are standardized (SMB: Supermom Beliefs as Beneficial; EF: Self-efficacy; WE: Work Engagement; PWB: Psychological Well-being; WEEK_HRS: Number of work hours in a week).

**Table 5. Alternate structural models.**

| Alternate Models | $\chi^2$ (df) | CMIN/df | $\Delta\chi^2$ ($\Delta$df) | RMSEA | SRMR | CFI |
|---|---|---|---|---|---|---|
| **Proposed Model** | 557.36 (380) | 1.47 | – | .039 | .058 | .944 |
| **Alternate Model** | 585.17 (381) | 1.54 | 27.81 (1) | .042 | .071 | .944 |

$N$ = 306. Indices with minimal difference have been represented up to 3 places of decimal Proposed model: Direct relation between SMB & EF, EF & WE, WE & PWB, and EF & PWB Alternate model: Direct relation between SMB & EF, EF & WE, and WE & PWB Df: Degrees of freedom; RMSEA: Root-mean-square error of approximation; SRMR: Standardized root-mean-square residual; CFI: Comparative fit index SMB Supermom (beneficial); EF: Self-efficacy; WE: Work Engagement; PWB: Psychological Well-being

**Table 6. Effects analysis using bootstrapping.**

| | Estimates | Bootstrap standard errors | BC percentile method (Lower Bound) | BC percentile method (Upper Bound) | $p$-value |
|---|---|---|---|---|---|
| **Direct Effects** | | | | | |
| **EF-PWB** | 0.392 | 0.106 | 0.184 | 0.609 | 0.002 |
| **EF-WE** | 0.493 | 0.071 | 0.353 | 0.626 | 0.001 |
| **WE-PWB** | 0.449 | 0.105 | 0.241 | 0.655 | 0.002 |
| **Indirect Effects** | | | | | |
| **EF-PWB** | 0.221 | 0.056 | 0.133 | 0.369 | 0.001 |
| **Total effects** | | | | | |
| **EF-PWB** | 0.614 | 0.074 | 0.476 | 0.759 | 0.001 |

$N$ = 306. All effects are standardised and represented up to 3 places of decimal. Two-tailed significance is reported. Bootstrap sample size: 1000; Bias-corrected: BC; BC confidence interval: 95% EF: Self-efficacy; WE: Work Engagement; PWB: Psychological Well-being

## Discussion

There is substantial skepticism or even cynicism about the notion of supermom and many have tried to resist the imposition of this belief. Therefore, the study aimed to explore how this belief may impact employed mothers' psychological well-being by affecting their self-efficacy and work engagement. This study can help get a deeper understanding of the impact of motherhood, the beliefs about the notion of supermom, and its implications for the psychological well-being of employed mothers. To our knowledge, this study attempts to be the first to decode the perceptions and beliefs about the notion of supermom and establish empirically its relationship with self-efficacy, work engagement, and psychological well-being. Thus, it makes a theoretical contribution to the interdisciplinary scholarship on motherhood, organizational behavior, and psychology [30,119–121].

The study employed an embedded/nested mixed methods research design that allowed the capturing of perceptions and beliefs about supermoms using symbolic interactionism as a framework for qualitative exploration of this notion. The quantitative part examines the relationship between supermom notion and self-efficacy, work engagement, and psychological well-being. This design not only allowed a nuanced understanding of different perceptions about the supermom notion through understanding the subjective meaning created by employed mothers about their social worlds but also enabled an empirical investigation of the relation of those perceptions with psychological and behavioral characteristics. This study design provides a valuable methodological contribution to the literature as we found no other studies utilizing such a research design to examine the impact of subjective perceptions on work engagement and psychological well-being through an empirical analysis [87,122,123].

Our sample consisted of mothers employed in urban areas, who were equivalently distributed in their beliefs about the supermom notion as beneficial or detrimental (who believed the

supermom notion to be a myth or a trap). Per our hypothesized model, positive perceptions about the supermom notion were significantly associated with self-efficacy. This self-efficacy was associated with higher psychological well-being, directly as well as through higher work engagement. This suggests that employed mothers in India, who perceive the supermom notion as beneficial are likely to feel competent and be engaged at work, and thus experience higher psychological well-being. The following sections discuss the theoretical implications of the individual findings.

Results of the study revealed that employed mothers who perceive the supermom notion to be beneficial have higher self-efficacy. This implies that belief of 'doing it all' makes them feel more efficacious in carrying out the various work and non-work activities. This finding contradicts the skeptical view of supermom [3] and the consequent understanding of its impact. However, the result mirrors the traditional Indian view of womanhood, in which despite India being patriarchal as a society, Indian women are worshiped as 'Shakti', a source of energy. This view considers them capable of multitasking, managing their families, earning for the house, and assisting significant others in many activities [124]. Previous research shows that when employed mothers perceive dual roles positively and do not consider them to be conflicting with each other, they will experience positive role balance and role ease [68]. Therefore, our findings suggest that an employed mother's positive notion of supermom reflects on the perceived benefits of creating a balance between these dual roles and is self-fulfilling as it increases their self-efficacy. The importance of supermom beliefs for self-efficacy, adds to the literature on motherhood and psychology [39,45,125].

The results further indicate how self-efficacy is associated with higher psychological well-being of employed mothers. This implies that the self-beliefs of employed mothers of being capable enough to carry out responsibilities in these dual roles improved their psychological well-being. This is supported by the Role Enhancement theory which discusses the usefulness of multiple roles in providing the benefits of social integration and a heightened sense of identity [126] to women, leading to higher well-being and the corollary of this would be well-being affected detrimentally by fewer roles [58]. A few studies have discussed the relationship between self-efficacy and the well-being of employed mothers [16,45], but they have not considered the mother's beliefs about the supermom notion. By highlighting the role of individual beliefs, the study adds to the limited literature on self-efficacy and psychological well-being of employed mothers and extends the motherhood, psychology, and mental health literature [29,44,120,121,127,128].

Besides, the results affirm that self-efficacy is associated with higher work engagement of employed mothers. This implies that the belief of employed mothers that they can accomplish work and non-work-related activities can relate to higher engagement in the work. Considering that the path analysis indicated a positive relation between a beneficial notion of supermom and self-efficacy (as covered above), the results also imply that this notion enhances work engagement by increasing self-efficacy [118]. This is consistent with the literature that discusses self-efficacy as a personal resource [71]. High personal resources lead to higher self-regard [85] and are helpful to individuals in developing a sense that they can successfully influence their environment [129]. This high self-regard consisting of high self-efficacy also leads to individuals having goal concordance, where individuals believe that there is a match between their goals and their capabilities, thus bringing higher levels of intrinsic motivation and satisfaction [130]. Literature shows that motivation and satisfaction are strongly associated with work engagement [131,132]. Although few studies have discussed the relationship between self-efficacy and work engagement in general [86,87], we have yet to find any literature discussing them in the context of employed mothers. Thus, this study adds to the interdisciplinary literature on motherhood and organizational behavior [9,22,133].

Further, the findings reveal that work engagement partially mediates the relation between self-efficacy and psychological well-being for employed mothers. This implies that self-efficacy can increase the psychological well-being of employed mothers directly and indirectly by improving their work engagement. Work engagement's positive relation with psychological well-being is supported by the fact that work engagement is understood as a positive motivational-affective state [86]. As implied by the Broaden and Build theory, positive emotional states can lead to the broadening of employees' momentary thought-action repertoires, as well as lead to personal, social, and psychological resources in the long term [134]. Therefore, through the experience of work engagement, employed mothers can enhance their psychological well-being. The relation between work engagement and psychological well-being is well-established in literature [18,88]. However, scholarly evidence on the relationship between self-efficacy, work engagement, and psychological well-being is minimal [89]. Also, we do not have relevant studies about employed mothers. By exploring these relationships, this study adds to the current motherhood, organizational behavior, psychology, and mental health literature [45,131].

The results show that work hours in a week did not impact the psychological well-being of employed mothers. A possible reason can be that mothers who perceive the supermom notion as beneficial, also have a strong sense of identity with their job. Even when they want to be the perfect mother to their children, they are also keen to be an ideal professional, who delivers as per or beyond expectations at the job. For such mothers, the advantage of tending to their children is offset by the disadvantage of not being able to give their best to the job. Studies have discussed the negative effect of part-time vis-à-vis a full-time job on the satisfaction of employed mothers [135].

The findings of our study may have some alternative explanations. Relationship between beliefs about supermom, self-efficacy work engagement, and psychological well-being can be influenced by various personal factors like upbringing, the economic and social conditions, respondent's perception about their own mother's capabilities, the influence of family and friends' perspective on a mother's role, self-identity and support system. It may also be affected by some organizational factors like leadership, supervisor and peer support, child support facilities, adequate training, conducive culture, absence of bias, and assured career growth.

## Practical implications for organizations

Organizations can play an important role in supporting employed mothers [136,137]. They can enhance mothers' work engagement and psychological well-being by taking certain initiatives that enable them to view the supermom notion more positively and increase their self-efficacy. First, they can organize or sponsor regular specialised workshops to develop mindfulness and a growth mindset amongst employed mothers (its possible effects are covered in the previous section). They can do regular check-ins to identify women who show symptoms of stress, disengagement, depression, and other mental health issues, and attend to them on a priority basis. Second, they can hold customized programs for mothers that guide them on how to take care of their mental and physical health and nutrition, manage their time, and lower their stress. These programs may also include peer-to-peer discussion forums where employed mothers feel free to share their challenges and offer solutions. Apart from equipping them with useful information, such programs can also make them feel valued and supported by the organization. They can also hire specialized counselors to guide and advise them as per requirement. Third, organizations can provide childcare and pediatrician-on-call facilities, flexible locations, and working hour policies to help employed mothers better integrate work and non-work lives, especially in the case of highly demanding jobs [138]. Fourth,

organizations can engage in coaching, mentoring, and role modeling, and utilise competency development to improve the self-efficacy of employed mothers, which is a malleable personal characteristic. Fifth, organizations can encourage supervisors to personally recognize, appreciate, and support mothers, and acknowledge the additional effort being put in by them to manage work and home. Sixth, organizations can invest in job design, competency mapping, and understanding individual goals, so that employees perceive work as meaningful and fulfilling.

## Limitations of the study

The study has certain limitations. First, convenience sampling restricted data collection to only a specific socio-economic class in urban India, where most employed mothers were from the middle to upper class, highly educated and well-paid and these aspects impacted the findings. Second, convenience sampling also could not ensure cultural diversity amongst the respondents stemming from city, state, social (e.g., caste), and religious background with which the respondents identified, which may influence the beliefs about the supermom notion. Besides, the sample was only from India which restricts the generalisability of the findings to the Indian context. Third, a cross-sectional study could not help assessing the long-term implications of perceiving supermom as a beneficial notion. This notion may be harmful to employed mothers, as it may cause stress and burnout over a period. Fourth, we could not incorporate aspects that could have directly affected the well-being of employed mothers, like living arrangements, support systems at home, and organizational work culture and employee support initiatives. Fifth, we could not capture demographic variables like age in actual figures. We captured them as categorical variables since our initial respondents hesitated to share their ages.

## Future recommendations

Future research can be based on a larger sample size of employed mothers, which is more representative of the diversity in India. For example, attention can be paid to including equivalent respondents with children in a variety of age groups, from diverse social, economic, and cultural backgrounds, and having different kinds of support systems at home and work. They may belong to dissimilar industries, different sizes of cities and organizations, and hold varied professional positions. Besides, such a study can be conducted across countries to understand cross-country and cross-cultural differences in supermom beliefs and their implication, and also examine the moderating effect of the cultural context. Furthermore, longitudinal studies can explore the long-term effects of such beliefs on the professional and personal lives of employed mothers, apart from their self-efficacy, work engagement, and mental health.

In future studies, it would also be useful to explore whether women who perceive being a supermom as a myth or trap may feel less efficacious, as grappling with daily challenges due to the dual roles of being a mother and an employee/worker, can lead to feelings of anxiety and stress [139]. Also, research can additionally consider aspects of the family environment such as family support, number of family members, and managerial support at organizations, as these can influence the self-efficacy, work engagement, and psychological well-being of employed mothers.

## Conclusion

The study shed a favorable light on the perceptions and beliefs about the notion of supermom which is considered as an epitome of motherhood indicating its positive relation with self-efficacy, work engagement, and psychological well-being. The qualitative questions uncovered beliefs and perceptions regarding the supermom notion in the Indian context. Considering

the scope of this paper, only two broad categories 'beneficial perception' and 'detrimental perception' were considered for a broad understanding. Future studies should focus on a layered understanding of the notion and its positive and negative impact on employed mothers' psychological well-being. The study revealed that employed mothers who perceive supermom as a beneficial notion have higher self-efficacy and these self-efficacious beliefs can lead to higher levels of work engagement, and eventually to higher psychological well-being. Some employed mothers believed that being a supermom is tiring at times, a trap, or a myth. But for other employed mothers, it served as an inspiration to aspire to be a supermom. Conclusively, believing in the supermom notion and finding it beneficial enhances the experience of motherhood and can be favorable for employed mothers' psychological well-being.

## Acknowledgments

The authors are grateful to UG3 students of AY 2023–24, Ms. Krisha Balsarkar, and Ms. Ananya Saxena of the FLAME University for their support with the literature search, and qualitative coding. We also thank Ms. Prachi Nawathe, Research Assistant, AY 2023-24 of theFLAME University for compiling the literature and managing the reference list.

## Author contributions

**Conceptualization:** Shalaka Sharad Shah, Shilpa Shinde.

**Data curation:** Shalaka Sharad Shah.

**Formal analysis:** Shalaka Sharad Shah, Smita Chaudhry.

**Methodology:** Shalaka Sharad Shah, Smita Chaudhry, Shilpa Shinde.

**Writing – original draft:** Shalaka Sharad Shah, Smita Chaudhry, Shilpa Shinde.

**Writing – review & editing:** Shalaka Sharad Shah, Smita Chaudhry, Shilpa Shinde.

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
