## [Decision Letter · Decision Letter 0]

30 Dec 2024

PONE-D-24-24738Supermoms - Tired, admired, or inspired? Decoding the impact of supermom beliefs: A study on Indian employed mothersPLOS ONE Dear Dr. Shah,Thank you for submitting your manuscript to PLOS ONE. After careful consideration, we feel that it has merit but does not fully meet PLOS ONE’s publication criteria as it currently stands. Therefore, we invite you to submit a revised version of the manuscript that addresses the points raised during the review process.

Following are few observations, which require considerate response:a. Literature review should include a section on theoretical rationale to justify the objectives and hypotheses formulated. b. Hypothesis 1 is not according to APA style.c. 'Supermom Belief' was measured qualitatively and then grouped into two themes as 'beneficial' or 'detrimental' whereas, path analysis only refers to beneficial category. It is not clear whether this type of data justifies use of path analysis. Similarly, pearson correlation has been applied to find out association between SMB and other variables.d. The whole article needs to be reviewed for grammatical mistakes.

We look forward to receiving your revised manuscript.

Kind regards,

Shazia Khalid, PhD

Academic Editor

PLOS ONE

Journal requirements: 1. When submitting your revision, we need you to address these additional requirements.Please ensure that your manuscript meets PLOS ONE's style requirements, including those for file naming. The PLOS ONE style templates can be found at https://journals.plos.org/plosone/s/file?id=wjVg/PLOSOne_formatting_sample_main_body.pdf and https://journals.plos.org/plosone/s/file?id=ba62/PLOSOne_formatting_sample_title_authors_affiliations.pdf 2. We note that you have indicated that there are restrictions to data sharing for this study. PLOS only allows data to be available upon request if there are legal or ethical restrictions on sharing data publicly. For more information on unacceptable data access restrictions, please see http://journals.plos.org/plosone/s/data-availability#loc-unacceptable-data-access-restrictions.  Before we proceed with your manuscript, please address the following prompts: a) If there are ethical or legal restrictions on sharing a de-identified data set, please explain them in detail (e.g., data contain potentially identifying or sensitive patient information, data are owned by a third-party organization, etc.) and who has imposed them (e.g., a Research Ethics Committee or Institutional Review Board, etc.). Please also provide contact information for a data access committee, ethics committee, or other institutional body to which data requests may be sent. b) If there are no restrictions, please upload the minimal anonymized data set necessary to replicate your study findings to a stable, public repository and provide us with the relevant URLs, DOIs, or accession numbers. For a list of recommended repositories, please seehttps://journals.plos.org/plosone/s/recommended-repositories. You also have the option of uploading the data as Supporting Information files, but we would recommend depositing data directly to a data repository if possible. We will update your Data Availability statement on your behalf to reflect the information you provide. 3. In the online submission form, you indicated that [Data cannot be shared publicly as it includes their email addresses. But Data can be made available from the Corresponding Author / Ethics Committee (contact via shalaka.shah@flame.edu.in) for researchers who meet the criteria for access to confidential data.]. All PLOS journals now require all data underlying the findings described in their manuscript to be freely available to other researchers, either 1. In a public repository, 2. Within the manuscript itself, or 3. Uploaded as supplementary information.This policy applies to all data except where public deposition would breach compliance with the protocol approved by your research ethics board. If your data cannot be made publicly available for ethical or legal reasons (e.g., public availability would compromise patient privacy), please explain your reasons on resubmission and your exemption request will be escalated for approval. 

Reviewers' comments:

Reviewer's Responses to Questions

**Comments to the Author**

1. Is the manuscript technically sound, and do the data support the conclusions?

Reviewer #1: Yes

Reviewer #2: Partly

2. Has the statistical analysis been performed appropriately and rigorously? 

Reviewer #1: Yes

Reviewer #2: Yes

3. Have the authors made all data underlying the findings in their manuscript fully available?

Reviewer #1: No

Reviewer #2: No

4. Is the manuscript presented in an intelligible fashion and written in standard English?

Reviewer #1: Yes

Reviewer #2: No

5. Review Comments to the Author

Reviewer #1: The article provides a valuable exploration of how societal ideals around motherhood affect the self-efficacy, work engagement, and psychological well-being of Indian employed mothers. Overall the study gave a comprehensive picture of the title but here are some recommendations that can improve the quality of the study.

1. The most important part that requires attention is the methodology of Qualitative part of study. Author must clearly state in detail about the methodological framework used to develop open ended questions.

2. The term 'supermom' is central to the study, but it lacks a precise and consistent definition across the article. While the qualitative responses attempt to capture different viewpoints, the notion remains somewhat vague. A clearer operational definition of what qualifies as 'supermom' behavior or expectations would help in interpreting the results.

3. considering and highlighting limitations and future recommendations would be a good addition to the article. The limitations in sampling, cultural diversity, and studying depth of negative consequences suggest room for improvement (Longitudinal study). Recommending these points in future research would enhance the scope and impact of the findings.

Reviewer #2: This study provides valuable insights into the psychological challenges faced by employed mothers in India, shedding light on the impact of "supermom" beliefs. The research is strong in many areas. However, there are gaps By addressing the following aspects, the authors could further strengthen their work

1. The entire document requires careful editing to correct grammatical errors and restructure overly long sentences for improved readability and flow.

2. Focus on the most critical aspects of the introduction, eliminating repetitive details to ensure a concise and engaging narrative.

3. The structural model is currently discussed in three different sections of the paper—under the "Measures" section, before hypothesis testing, and after the results. To improve clarity and coherence, it would be more effective to integrate these discussions into a single, unified section, ensuring a smooth flow of information and avoiding redundancy.

4. The authors have taken an important step by conducting EFA and reporting reliability, the absence of methodological details, validity metrics, and contextual interpretations weakens the transparency and robustness of the findings. Addressing these gaps will enhance the credibility and impact of the analysis.

5. Provide additional details about the path analysis process, particularly addressing the use of the "Supermom Beliefs as Beneficial" variable, which was nominal and not appropriate for Pearson correlation.

6. Include bootstrapped estimates of indirect effects and confidence intervals to substantiate claims regarding mediation.

7. The significant chi-square statistic suggests a potential misfit between the model and the data. While this is common in large samples, the authors should explicitly discuss the limitations of the chi-square test and justify reliance on alternative fit indices.

8. Tie the findings to established psychological theories, such as self-efficacy theory or theories of work engagement, to provide a more nuanced explanation of the results.

9. Discuss how these findings can inform interventions for employed mothers, including workplace policies and mental health initiatives.

10. Consider potential confounders or alternative explanations for the observed relationships, and explicitly acknowledge the limitations of the study design.

6. PLOS authors have the option to publish the peer review history of their article (what does this mean? ). If published, this will include your full peer review and any attached files.

**Do you want your identity to be public for this peer review?** For information about this choice, including consent withdrawal, please see our Privacy Policy .

Reviewer #1: No

Reviewer #2: No

---

## [Author Response · Author response to Decision Letter 1]

9 Feb 2025

We have uploaded a separate rebuttal letter named as 'Response to Reviewers'.

---

## [Editor Report · Decision Letter 1]

10 Mar 2025

Supermoms - Tired, admired, or inspired? Decoding the impact of supermom beliefs: A study on Indian employed mothers

PONE-D-24-24738R1

Dear Author,

We’re pleased to inform you that your manuscript has been judged scientifically suitable for publication and will be formally accepted for publication once it meets all outstanding technical requirements.

Kind regards,

Shazia Khalid, PhD

Academic Editor

PLOS ONE
---

## [Editor Report · Acceptance letter]

PONE-D-24-24738R1

PLOS ONE

Dear Dr. Shah,

I'm pleased to inform you that your manuscript has been deemed suitable for publication in PLOS ONE. Congratulations! Your manuscript is now being handed over to our production team.

Kind regards,

on behalf of

Professor Shazia Khalid

Academic Editor

PLOS ONE